# Vegetal-FRCM Failure under Partial Interaction Mechanism

**Virginia Mendizabal *** , **Borja Martínez** , **Luis Mercedes, Ernest Bernat-Maso** and **Lluis Gil**

Department of Strength of Materials and Structural Engineering, School of Industrial, Aerospace and Audiovisual of Terrassa, Universitat Politècnica de Catalunya, C/Colom 11, TR45, 08222 Terrassa, Spain
* Correspondence: virginia.dolores.mendizabal@upc.edu

**Featured Application: The reinforcement of concrete structures using FRCM is a rather novelty approach that overcomes some of the drawbacks of organic composites such as carbon fiber CFRP. Moreover, the use of vegetal fabrics has a direct effect on the carbon footprint. Some calculations deal with a full composite action between the matrix and the fibers. Nevertheless, it has been noticed that sometimes the chemical and mechanical adherence between the cementitious matrix and the vegetal fabric does not perform as a perfect bond, and it is necessary to take into account that the matrix–fiber bond is weak. This research provides insight into the limits of the strength for these situations. Partial interaction allows the composite to strengthen with larger deformation while maintaining an interesting ultimate capacity. The characterization of this type of performance may help to design resilient solutions for concrete structures under critical bonding situations produced by ambient or manufacturing reasons.**

**Abstract:** FRCM is a strengthening system based on composite material made of a cementitious matrix and fabrics. This strengthening system has been studied and researched, obtaining analytical predictive models where it is common to assume full composite action between components. Through using non-typical materials for these composites, it has been seen that, in some cases, the previous assumption cannot be taken. In this situation, traditional analytical models such as ACK or tri-linear ones do not offer a reasonable prediction. This work researches the behavior of synthetic and naturally coated vegetal-FRCM with partial interaction through the characterization of the materials through tensile tests. Yarns, meshes and different FRCM coupons were manufactured and mechanically tested using different types of coatings and fabrics. The use of colophony and Arabic gum as natural coatings provided similar mechanical properties to the cotton and hemp yarns and meshes conformed. Partial interaction was found when using epoxy as a natural resin to coat the reinforcement while maintaining the mechanical properties in the same order of magnitude. A new two-stage model is proposed to fit stress–strain mechanical test, and it is reliable and accurate for cotton specimens.

**Keywords:** FRCM; vegetal fibers; coating; fiber–matrix bond; mechanical testing

## 1. Introduction

At present, society's awareness of sustainability is rising. More and more environmentally friendly solutions are researched and demanded by all types of industries. The construction industry is not an exception. The research towards sustainability has already started, especially in the field of strengthening structures. In particular, fiber-reinforced cementitious matrix (FRCM) has been studied, replacing with vegetal fibers such as hemp, cotton or sisal [1–3] the synthetic and high-properties fibers such as glass, carbon or basalt [4], and results are meaningful. Mercedes et al. [2] compared the mechanical properties of four different FRCMs, with fibers coated with epoxy and polyester resins, where the tensions and multi-cracking failure reached, suggesting that their properties are comparable to the synthetic-fiber FRCMs. The research on the topic of coatings is still in development.

The coating of the fiber is needed to give the yarns a mesh consistency when they are extracted from the mold to protect the fibers from the alkaline environment of the fresh

mortar and also to improve the interphase fiber–matrix [5]. Donnini et al. [6] tested the difference in the interphase of a carbon yarn embedded into a mortar matrix, with and without coating. In the post-elastic phase, when the fiber is debonding from the matrix, uncoated yarns present a brittle failure, while impregnated yarns present a more ductile one.

The coatings used for strengthening structures are usually synthetic ones [7]. Fernandes et al. [8] conducted a review of vegetal fibers in polymeric composites, and the most common polymers used are epoxy, polyester (PES) and polypropylene (PP) resins. The research on more sustainable coating is limited. Mostly chemical treatments are applied to the fibers, such as silane or alkaline treatments [8,9]. Moreover, plasma application was studied to vary the properties of the fiber [10], increasing its mechanical properties tested through a tensile test of a non-woven sample.

More recently, new coatings and treatments are being developed. Abbas et al. [11] used beeswax emulsion to coat vegetal fibers, Bakhtiari Ghaleh et al. [12] coated the fibers with resin and micro-silica and Veloso de Carvalho et al. [13] coated curaua fibers with polyaniline and magnetite to improve the interphase with the cement matrix. Zhao et al. [14] coated sisal fibers with graphene oxide to mitigate the degradation process. The interest of the field of science to increase the knowledge of coatings for structural application and the need for more sustainable materials were crucial when deciding the coatings for this research. Finally, completely natural coatings were selected to be compared with the performance of synthetic ones, specifically colophony and Arabic gum.

Colophony [15] is a pine resin used in the manufacturing of adhesives, coatings or tints, and Arabic gum is a resin used in the alimentary and pharmaceutical industry to give flexibility to the products [16]. Depending on the level of flexibility required, the concentration of the Arabic gum varies between 46 and 85%.

In order to test the mechanical properties of FRCM samples, the tensile test is one of the most performed. There are different gripping methods used: clamps, which apply compression to the subjected ends, anchoring it [4,17], and the clevis grip method, which uses two metallic plates on each end to transmit the load by surface shear [17,18].

In order to model the behavior of FRCM, common models used for tensile behavior consist of the ACK model [4,19] or the simplified tri-linear model [4]. These models consider three stages, with the law of mixtures in the first stage until the first crack appears, constant stress until the mortar is completely cracked, and a third stage that considers that all the load is bore by the fibers. However, it has been proved that for the first and second stages, ACK can overestimate the strength of the specimens [20]. Moreover, the second and third stage's slopes can be similar [21], and new models have been proposed [22]. Sometimes the material might not perform with full interaction between components from the very beginning. This would be the case in which the coating of the fibers does not guarantee a perfect bonding, and materials tend to slip into each other, producing a different type of failure [23].

This work aims to increase the knowledge about FRCM performance spanning the analysis towards a partial interaction. Therefore, an experimental work has been developed to compare the performance of different synthetic-coated and natural-coated vegetal FRCMs, through the whole characterization of the materials: from the yarn to the final FRCM composite material. A two-stage analytical model is also proposed and discussed.

## 2. Materials and Methods

### 2.1. Materials

The materials selected for this study were as follows: Rombull[TM] (Alicante, Spain) vegetal yarns, Ø 1.5 mm cotton plaited yarns and Ø 0.5 mm hemp spun yarns. In order to coat the fibers, 2 synthetic and 2 natural coatings were chosen: MasterBrace[TM] P3500 (from BASF, Ludwigshafen, Germany) epoxy resin (due to its high adhesion and low viscosity), SilmarTM SIL66BQ-249A (from VIRALSURF, Biarritz, France) polyester resin (due to its flexibility), Arabic gum and colophony. For natural coatings, it was decided to study two

different cases for each one. For colophony, it was decided to dissolve it in two different solvents: acetone and turpentine [24]. For Arabic gum that was dissolved in distilled water at high temperature, it was decided to prepare it with and without 15 min of ultrasound application. Ultrasounds make the resin more homogeneous through cavitation [25]. The matrix consisted of Sika[TM] Monotop-612 mortar (Baar, Switzerland), with a variation on the water content of 16.6% instead of the 14.5% recommended by the manufacturer) to facilitate the FRCM manufacturing process according to previous research [2].

The materials' properties can be found in Tables 1–3.

**Table 1.** Yarns properties.

| Properties | Hemp | Cotton |
|---|---|---|
| Yarn diameter [mm] [1] | 0.5 | 1.5 |
| Number of yarns/tuft | 8 | 3 |
| Maximum load [N] [1] | 83.4 | 78.5 |

[1] Manufacturer data.

**Table 2.** Synthetic resin properties.

| Properties | MasterBrace P 3500 (EP) | SILMAR SIL66BQ-249A (PES) |
|---|---|---|
| Density [g/cm$^3$] [1] | 1.05 | - |
| Tensile strength [MPa] [1] | $22.9 \pm 4$ | 69.4 |
| Elongation [%] [1] | $18.2 \pm 7$ | 1.9 |
| Flexural strength [MPa] [1] | No break | 138.6 |
| Flexural modulus [MPa] [1] | 233.1 | 4095 |

[1] Provided by each supplier.

**Table 3.** Mortar properties.

| Properties | Sika Monotop-612 |
|---|---|
| Chemical composition [1] | Prepared cement mortar, improved with synthetic resins and silica fume, and reinforced with polyamide fibers |
| Density of fresh mortar [1] | 2.1 kg/L (at 20 °C) |
| Granulometry [1] | 0–2 mm |
| Flexural strength [2] | 6.79 MPa |
| Tensile strength [2] | 2.26 MPa |

[1] Provided by supplier. [2] Obtained through the testing following the normative [26].

### 2.2. Methods

The methodology consisted of firstly testing all possible combinations of yarns and coating through a yarn tensile test. By analyzing the results presented in Section 4.1. for the natural coatings, it was decided to continue the experimental campaign with the combinations that show the best mechanical improvement for each fiber. The following experimental campaign consisted of the tensile test of meshes and also of the FRCM of each combination selected.

### 2.2.1. Preparation of the Yarns

The preparation of the yarns consisted of cutting 40 cm lengths of both types of yarns. In the case that the yarns had to be coated, they were allocated in a wooden loom and coated manually using a brush following Table 4 to know the concentrations of the different resins. The yarns were left for 48 h to reticulate the resins and extracted from the loom. Three specimens per case were manufactured.

**Table 4.** Coating types and preparation.

| Coating | Preparation | Reference |
|---|---|---|
| Epoxy (EP) | 39 g PartA/11 g PartB | Manufacturer |
| Polyester (PES) | Resin with a 2% mass of the accelerating agent | Manufacturer |
| Arabic gum (GA) | 40% mass dissolved in distilled water | Ref. [16] Choosing the more flexible concentration. |
| Arabic gum with ultrasound (GAU) | 40% GA dissolved in distilled water, with 15 min ultrasound | Application of ultrasounds to reduce the air inside the solution, homogenizing it [25] |
| Colophony + turpentine (COLTUR) | 1:6 relation dissolved in turpentine | Ref. [24] Low concentration due to the brittle behavior of the resin |
| Colophony + acetone (COLAC) | 1:6 relation dissolved in acetone | Ref. [24] Acetone as a substitute for turpentine. |

Once the resins were reticulated, a glass FRP was set on both ends to improve the transmission of the load between the test equipment and the specimens and to avoid local stress concentration during the test. (Figure 1), as it was previously performed in [2]. After 48 h, the specimens were ready to be tested.

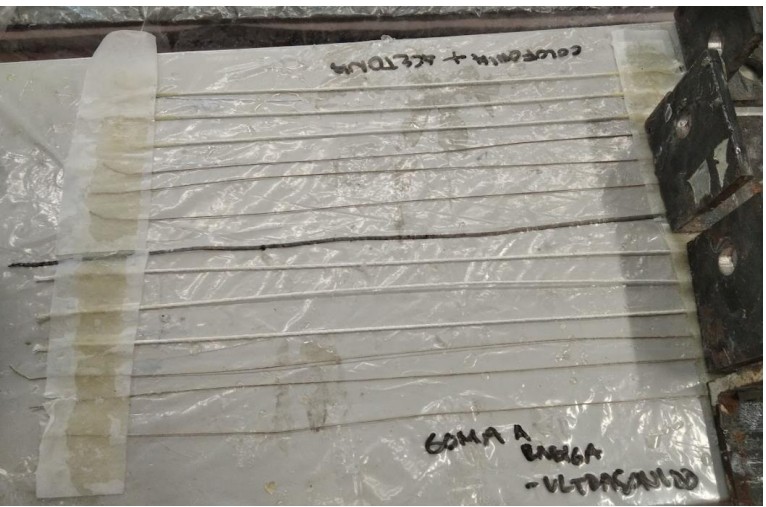

**Figure 1.** Preparation of the yarns.

After testing all the combinations, only four coatings per fiber were selected: epoxy and polyester resin, Arabic gum without ultrasounds and colophony dissolved in turpentine for cotton fiber, and colophony dissolved in acetone for hemp. The details that support this election are explained in Sections 3.1 and 4.1.

2.2.2. Preparation of the Meshes

The meshes consisted of 4 tufts of 40 cm in length, each 25 mm a weave crossed the tufts creating the mesh. The number of yarns per tuft was different for each fiber: hemp consisted of 8 yarns/tuft and cotton 3 yarns/tuft (Figure 2).

After weaving, they were coated following the concentrations detailed in Table 4, with the difference that each fiber was coated with four types of coatings: epoxy and polyester resin, Arabic gum, colophony dissolved in turpentine for cotton, and colophony dissolved in acetone for hemp. By using a mold of 30 cm × 40 cm, a dosage of 100 g of coating for hemp meshes was used, and 150 g was used for cotton meshes.

After 48 h of reticulating for synthetic resins and 120 h for natural coatings, in the environmental conditions of the laboratory, the meshes were extracted from the loom, and a glass-FRP was set at both ends to improve the load transmission between the test equipment and the specimens. After another 48 h, the specimens were ready to be tested.

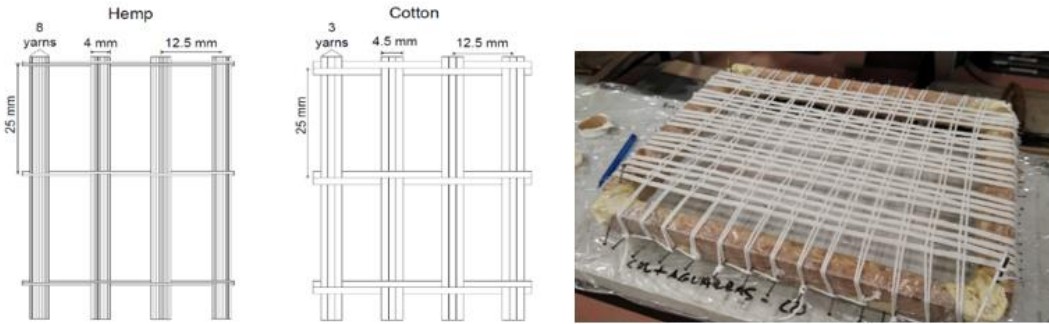

**Figure 2.** Meshes geometry (**left**), wooden mesh (**right**).

### 2.2.3. Preparation of the FRCM

The size of FRCM specimens was $50 \times 400 \times 10$ m [2]. The preparation for FRCM started with the preparation of the meshes, following the exact procedure explained in the previous section, but the glass-FRP endings were not executed. Moreover, a wooden mold, where FRCM would be manufactured, was prepared. It consisted of a wooden plate with pieces that defined the required geometry. Each mold was covered with plastic tape and coated with vaseline to make the extraction easier.

The procedure consisted of the application of a 5 mm layer of mortar, allocating the mesh and pressing against the mortar to make sure that it penetrates through the holes of the mesh, and the addition of another 5 mm layer of mortar, leveling the surface. After 28 days of curing, four metallic plates were bonded to the ends of the specimens with high-strength adhesive (Loctite$^{\text{TM}}$ EA 3425, Henkel Iberica SA, Barcelona, Spain) to allow the load to be applied through a Clevis grip method [27]. After 48 h of hardening, the specimens were ready to be tested.

### 2.2.4. Test Set-Ups

#### Yarns and Meshes Tensile Test

Tensile tests followed the code EN ISO 13934-1/2 [28], but they were adapted to the particular requirements. It consisted of the use of an electromechanical press, MTS Insight 10 kN range, with an extensometer of 25 mm range to measure the deformation of the specimen. The equipment–specimen grip method changed depending on the specimen tested, and also the extensometer–specimen grip.

For the yarn's tensile test, small clamps were used to hold the specimen, and the extensometer was placed directly against the fiber using rubber pieces to fix it (Figure 3, left). In the case of meshes, big clamps were used with 5 cm width, and the extensometer was placed against an FRP in contact with a rubber to maintain the position (Figure 3, middle).

The development of the test consisted of a displacement-controlled speed of 5 mm/min, applying a pre-load of 5 N to ensure the specimen was in tension. Once the extensometer was placed, the tests started until the specimen failed through rupture or slippage [2].

#### FRCM Tensile Test

FRCM testing procedure took as a reference the AC434-0213-R1 procedure [29]. A clevis grip method was used, and the extensometer was subjected using two metallic L-shaped plates that were subjected against the specimens using magnets on the metallic plates (Figure 3, right).

The configuration used was the same used in yarns and meshes.

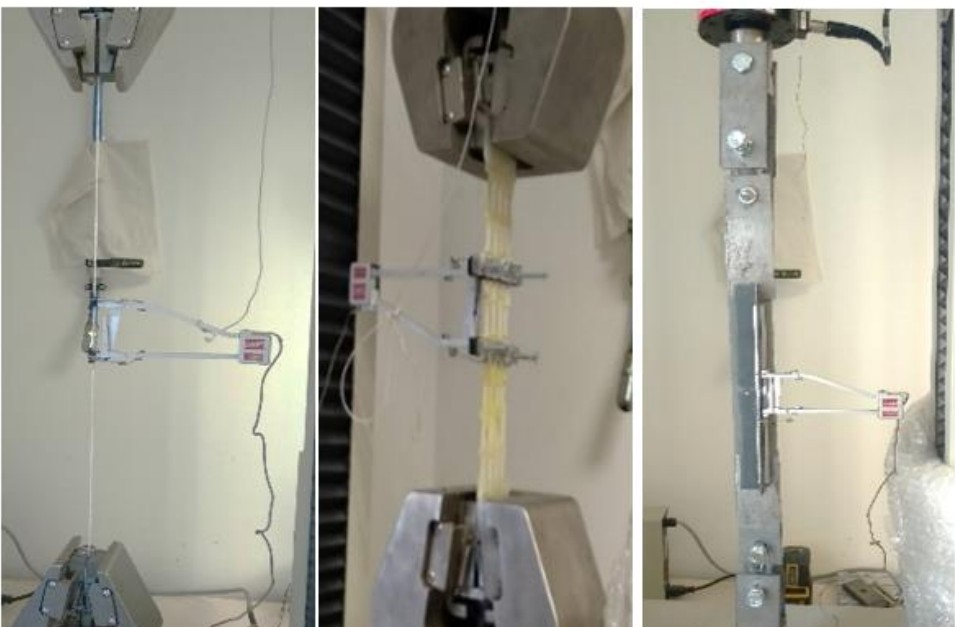

**Figure 3.** Tensile test setup: yarns (**left**), meshes (**middle**), FRCM (**right**).

## 3. Results

### 3.1. Yarns Tensile Test

The results of the yarn's tensile test can be found in Tables 5 and 6, where the maximum load ($F_{peak}$) and peak displacement of the extensometer ($\delta_{peak}$) are detailed per specimen. Additionally, the maximum load, peak displacement of the extensometer, maximum strength ($\sigma_{peak}$), strain at peak ($\varepsilon_{peak}$) and Young's modulus (E) per case are detailed. Moreover, Young's modulus was calculated using a linear regression of the σ-ε curve. In order to calculate the strength, the cross-section of the yarn was used (1.76 mm$^2$ for cotton and 0.20 mm$^2$ for hemp). For the strain, the displacement was divided by the natural longitude of the extensometer (50 mm).

Both yarns were tested with all the possible coatings to select the better-performing ones for further research. Figures 4 and 5 show the average curves of the test for each yarn. The three repetition per case curves can be found on Supplementary materials in Figures S1–S14.

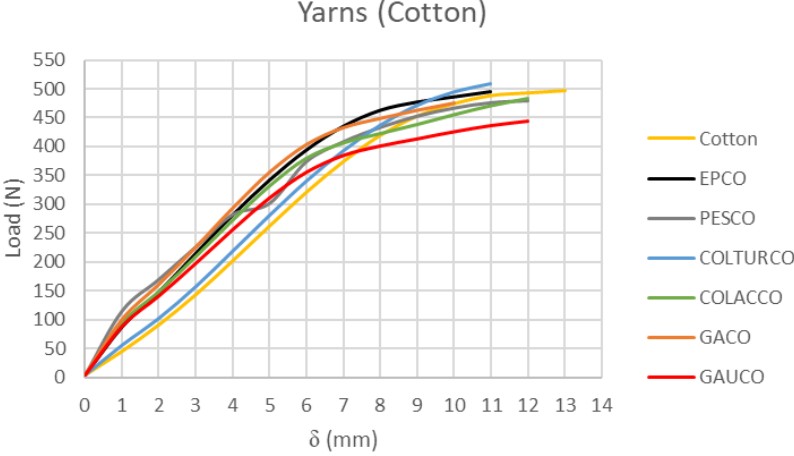

**Figure 4.** Load–extensometer curves for cotton yarns.

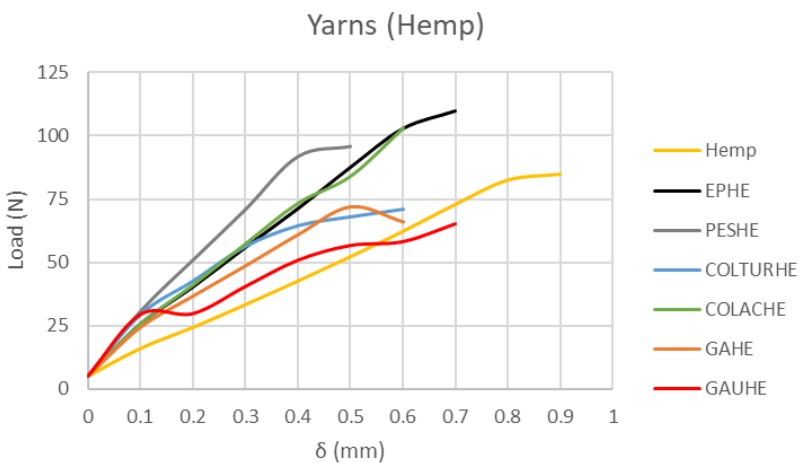

**Figure 5.** Load–extensometer curves for hemp yarns.

**Table 5.** Cotton yarns' tensile test results.

| Fiber | Coating | F$_{peak}$ (N) | δ$_{peak}$ (mm) | Average per Case | | | | |
| --- | --- | --- | --- | --- | --- | --- | --- | --- |
| | | | | F$_{peak}$ (N) | δ$_{peak}$ (mm) | σ$_{peak}$ (MPa) | ε$_{peak}$ (%) | E (MPa) |
| Cotton (CO) | - | 498 | 11.33 | 489 (4%) | 11.22 (13%) | 276 | 22.45 | 1590 |
| | | 499 | 12.68 | | | | | |
| | | 470 | 9.74 | | | | | |
| | EP | 496 | 11.03 | 486 (1%) | 9.85 (12%) | 275 | 19.69 | 1701 |
| | | 476 | 8.57 | | | | | |
| | | 486 | 9.78 | | | | | |
| | PES | 479 | 11.43 | 477 (2%) | 11.43 (4%) | 270 | 22.87 | 1406 |
| | | 479 | 11.89 | | | | | |
| | | 474 | 10.99 | | | | | |
| | GA | 479 | 10.31 | 457 (8%) | 8.97 (26%) | 258 | 18.24 | 1829 |
| | | 477 | 10.27 | | | | | |
| | | 412 | 6.34 | | | | | |
| | GAU | 449 | 12.59 | 446 (1%) | 12.05 (3%) | 253 | 24.10 | 1589 |
| | | 444 | 11.84 | | | | | |
| | | 446 | 12.51 | | | | | |
| | COLTUR | 515 | 11.43 | 513 (1%) | 11.51 (3%) | 290 | 23.02 | 1649 |
| | | 511 | 11.05 | | | | | |
| | | 513 | 11.83 | | | | | |
| | COLAC | 484 | 11.81 | 467 (4%) | 10.94 (12%) | 264 | 21.88 | 1710 |
| | | 445 | 9.38 | | | | | |
| | | 472 | 11.34 | | | | | |

**Table 6.** Hemp yarns' tensile test results.

| Fiber | Coating | $F_{peak}$ (N) | $\delta_{peak}$ (mm) | Average per Case | | | | |
|---|---|---|---|---|---|---|---|---|
| | | | | $F_{peak}$ (N) | $\delta_{peak}$ (mm) | $\sigma_{peak}$ (MPa) | $\varepsilon_{peak}$ (%) | E (MPa) |
| Hemp (HE) | - | 91 | 0.94 | 89 (5%) | 0.90 (11%) | 454 | 1.79 | 24,160 |
| | | 84 | 0.78 | | | | | |
| | | 93 | 0.90 | | | | | |
| | EP | 110 | 0.59 | 110 32%) | 0.63 (10%) | 560 | 1.26 | 41,252 |
| | | 114 | 0.72 | | | | | |
| | | 106 | 0.56 | | | | | |
| | PES | 103 | 0.42 | 111 (9%) | 0.51 (20%) | 585 | 1.10 | 51,561 |
| | | 107 | 0.43 | | | | | |
| | | 123 | 0.67 | | | | | |
| | GA | 94 | 0.50 | 76 (20%) | 0.54 (13%) | 387 | 0.90 | 31,013 |
| | | 68 | 0.61 | | | | | |
| | | 66 | 0.50 | | | | | |
| | GAU | 67 | 0.47 | 69 (4%) | 0.61 (26%) | 350 | 1.21 | 26,724 |
| | | 70 | 0.56 | | | | | |
| | | 69 | 0.76 | | | | | |
| | COLTUR | 71 | 0.54 | 71 (2%) | 0.45 (26%) | 360 | 0.91 | 34,364 |
| | | 71 | 0.42 | | | | | |
| | | 70 | 0.32 | | | | | |
| | COLAC | 96 | 0.47 | 95 (12%) | 0.55 (13%) | 486 | 1.09 | 40,921 |
| | | 105 | 0.61 | | | | | |
| | | 85 | 0.55 | | | | | |

The average curves were calculated as the average of the load per displacement.

In order to continue with the experimental campaign, from the vegetal coatings that had two possibilities each, it was selected the one that showed the best performance on the yarns. Arabic gum without the ultrasounds because it decreased the maximum load, and in the case of colophony, dissolved in turpentine for cotton and dissolved in acetone for hemp yarns, as they were the ones that showed a higher maximum load.

### 3.2. Meshes Tensile Test

The results of the yarn meshes' tensile test can be found in Tables 7 and 8, where the average maximum load, peak displacement, maximum strength and strain at the peak are detailed. Moreover, Young's modulus was calculated using linear regression. Both yarns were tested with all the possible coatings to select the ones for further research, where the maximum load ($F_{peak}$) and peak displacement of the extensometer ($\delta_{peak}$) are detailed per specimen. Additionally, the maximum load, peak displacement of the extensometer, maximum strength ($\sigma_{peak}$), strain at peak ($\varepsilon_{peak}$) and Young's modulus (E) per case are detailed. Moreover, Young's modulus was calculated using a linear regression of the σ-ε curve. In order to calculate the strength, the cross-section of the yarn was used (21.20 mm² for cotton and 6.28 mm² for hemp). For the strain, the displacement was divided by the natural longitude of the extensometer (50 mm). Figures 6 and 7 show the average curves of the test for each yarn. The three repetition per case curves can be found on Supplementary materials in Figures S15–S22.

**Table 7.** Cotton meshes' tensile test results.

| Fiber | Coating | $F_{peak}$ (N) | $\delta_{peak}$ (mm) | Average per Case | | | | |
| --- | --- | --- | --- | --- | --- | --- | --- | --- |
| | | | | $F_{peak}$ (N) | $\delta_{peak}$ (mm) | $\sigma_{peak}$ (MPa) | $\varepsilon_{peak}$ (%) | E (MPa) |
| Cotton (CO) | EP | 5816 | 11.26 | 5523 (4%) | 10.79 (11%) | 260.43 | 21.58 | 1371 |
| | | 5198 | 10.48 | | | | | |
| | | 5565 | 10.65 | | | | | |
| | PES | 6165 | 10.07 | 6118 (1%) | 10.71 (5%) | 288.50 | 21.41 | 1301 |
| | | 6145 | 10.93 | | | | | |
| | | 6046 | 11.15 | | | | | |
| | GA | 4966 | 13.82 | 4974 (7%) | 13.23 (11%) | 234.13 | 29.15 | 958 |
| | | 4919 | 13.01 | | | | | |
| | | 5010 | 16.22 | | | | | |
| | COLTUR | 4987 | 7.36 | 5491 (8%) | 8.86 (15%) | 258.94 | 17.71 | 1500 |
| | | 5502 | 9.63 | | | | | |
| | | 5984 | 9.57 | | | | | |

**Table 8.** Hemp meshes' tensile test results.

| Fiber | Coating | $F_{peak}$ (N) | $\delta_{peak}$ (mm) | Average per Case | | | | |
| --- | --- | --- | --- | --- | --- | --- | --- | --- |
| | | | | $F_{peak}$ (N) | $\delta_{peak}$ (mm) | $\sigma_{peak}$ (MPa) | $\varepsilon_{peak}$ (%) | E (MPa) |
| Hemp (HE) | EP | 3622 | 0.89 | 4712 (16%) | 0.96 (6%) | 750.32 | 1.93 | 40,535 |
| | | 5161 | 0.99 | | | | | |
| | | 5354 | 1.00 | | | | | |
| | PES | 3838 | 1.03 | 3575 (12%) | 0.60 (32%) | 548.25 | 1.20 | 45,589 |
| | | 3729 | 0.60 | | | | | |
| | | 3158 | 0.64 | | | | | |
| | GA | 2422 | 0.75 | 2507 (7%) | 1.14 (40%) | 399.20 | 2.27 | 32,904 |
| | | 2456 | 1.76 | | | | | |
| | | 2644 | 0.88 | | | | | |
| | COLAC | 2611 | 0.54 | 2583 (1%) | 0.65 (14%) | 411.30 | 1.29 | 44,306 |
| | | 2586 | 0.71 | | | | | |
| | | 2553 | 0.68 | | | | | |

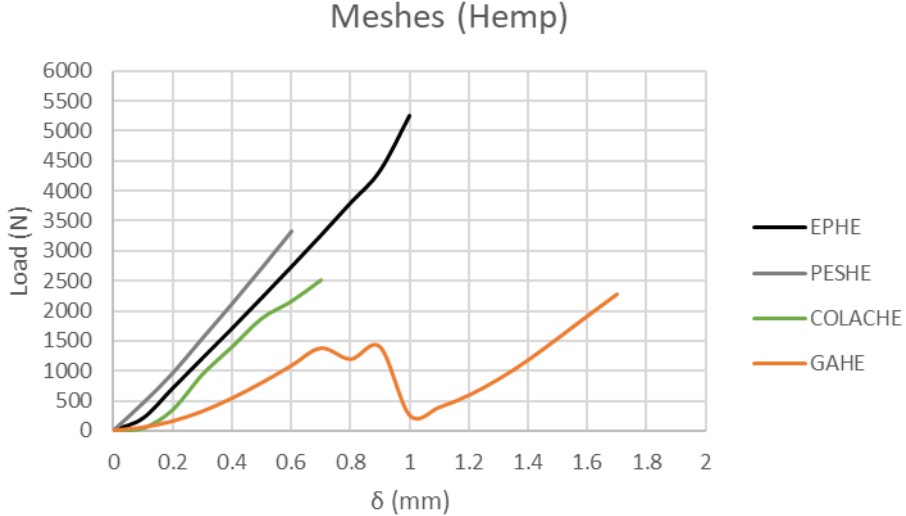

**Figure 6.** Load–Extensometer curves for hemp meshes.

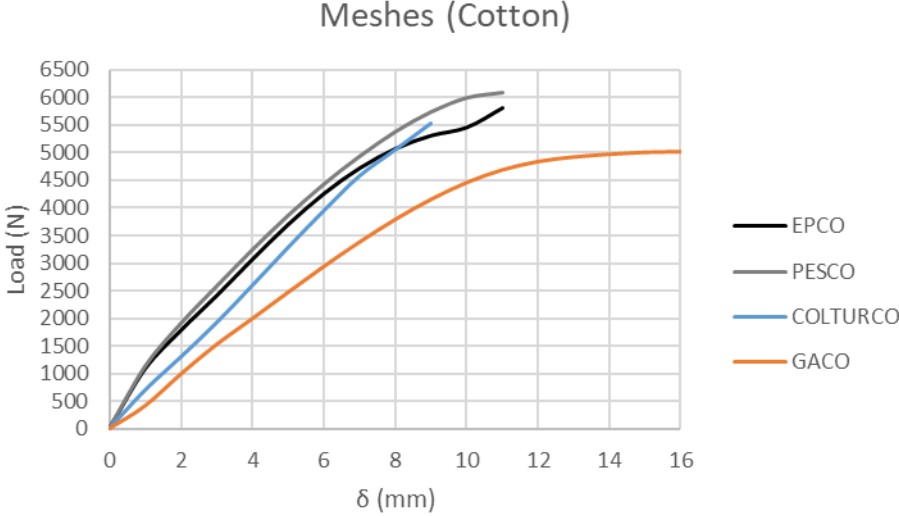

**Figure 7.** Load–Extensometer curves for cotton meshes.

The average curves have been calculated as the average of the load per displacement.

### 3.3. FRCM Tensile Test

FRCM tensile test results can be found in Tables 9 and 10, where the maximum load ($F_{peak}$) and peak displacement of the extensometer ($\delta_{peak}$) are detailed per specimen. Additionally, the maximum load, peak displacement of the extensometer, maximum strength ($\sigma_{peak}$) and strain at peak ($\varepsilon_{peak}$) are detailed per case. The strength is calculated using the cross-section of the mesh (21.20 mm$^2$ for cotton and 6.28 mm$^2$ for hemp), as it is considered that during the failure, the mortar is no longer contributing to bearing the load.

Figure 8 shows the three failure modes found. The types of failure are defined as follows: (I) shear failure of the contact plane; (II) Rupture of the mesh; (III) internal sliding of the mesh from the matrix.

Figures 9 and 10 correspond to the average load–extensometer curves of each case. The three repetition per case curves can be found on Supplementary materials in Figures S23–S30.

The average curves have been calculated as the average of the load per displacement.

**Table 9.** Cotton FRCM's tensile test results.

| Fiber | Coating | $F_{peak}$ (N) | $\delta_{peak}$ (mm) | Average per Case | | | | |
| | | | | $F_{peak}$ (N) | $\delta_{peak}$ (mm) | $\sigma_{mesh,peak}$ (MPa) | $\varepsilon_{peak}$ (%) | Failure |
|---|---|---|---|---|---|---|---|---|
| Cotton (CO) | EP | 2913 | 17.81 | 3562 (10%) | 20.68 (13%) | 118 | 41.36 | I |
| | | 2931 | 20.31 | | | | | |
| | | 3566 | 23.93 | | | | | |
| | PES | 3091 | 17.11 | 2687 (1%) | 18.02 (7%) | 147 | 36.04 | I |
| | | 3146 | 18.85 | | | | | |
| | | - | - | | | | | |
| | GA | 2004 | 23.67 | 2882 (15%) | 23.66 (1%) | 104 | 47.32 | III |
| | | 2532 | 25.00 | | | | | |
| | | 1787 | 20.49 | | | | | |
| | COLTUR | - | - | 3406 (3%) | 20.51 (1%) | 160 | 41.02 | III |
| | | 3126 | 25.00 | | | | | |
| | | 3246 | 25.00 | | | | | |

**Table 10.** Hemp FRCM's tensile test results.

| Fiber | Coating | $F_{peak}$ (N) | $\delta_{peak}$ (mm) | Average per Case | | | | |
| | | | | $F_{peak}$ (N) | $\delta_{peak}$ (mm) | $\sigma_{mesh,peak}$ (MPa) | $\varepsilon_{peak}$ (%) | Failure |
|---|---|---|---|---|---|---|---|---|
| Hemp (HE) | EP | 2198 | 3.30 | 2294 (5%) | 3.30 (13%) | 399 | 6.60 | I |
| | | 2194 | 3.00 | | | | | |
| | | 2490 | 3.90 | | | | | |
| | PES | 2828 | 5.15 | 2768 (9%) | 5.00 (17%) | 421 | 10.00 | II |
| | | 2538 | 6.35 | | | | | |
| | | 2939 | 3.51 | | | | | |
| | GA | 1896 | 25.00 | 1857 (8%) | 24.71 (1%) | 295 | 49.43 | III |
| | | 1728 | 24.15 | | | | | |
| | | 1837 | 25.00 | | | | | |
| | COLAC | - | - | 1568 (13%) | 25 (12%) | 249 | 50.00 | III |
| | | 1428 | 25.00 | | | | | |
| | | 1707 | 25.00 | | | | | |

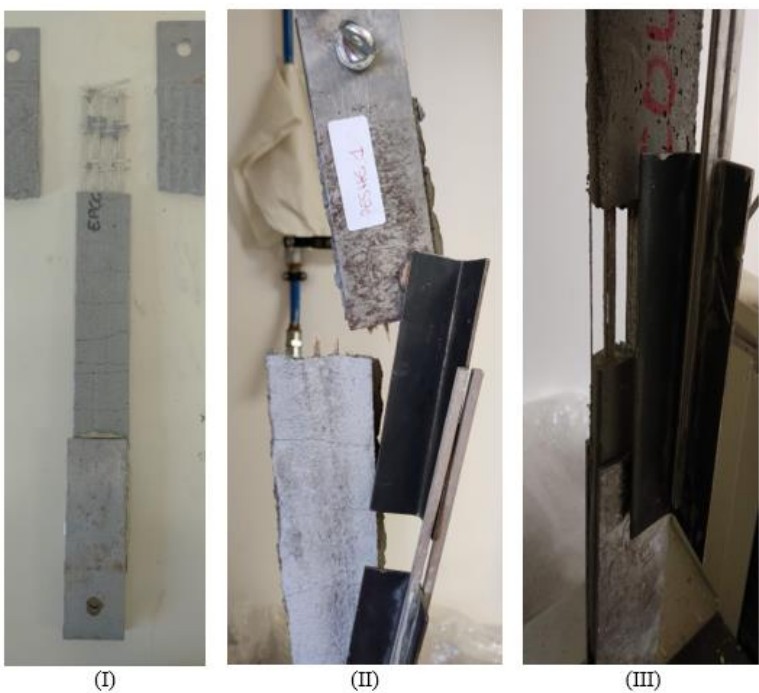

**Figure 8.** Failure modes of FRCM.

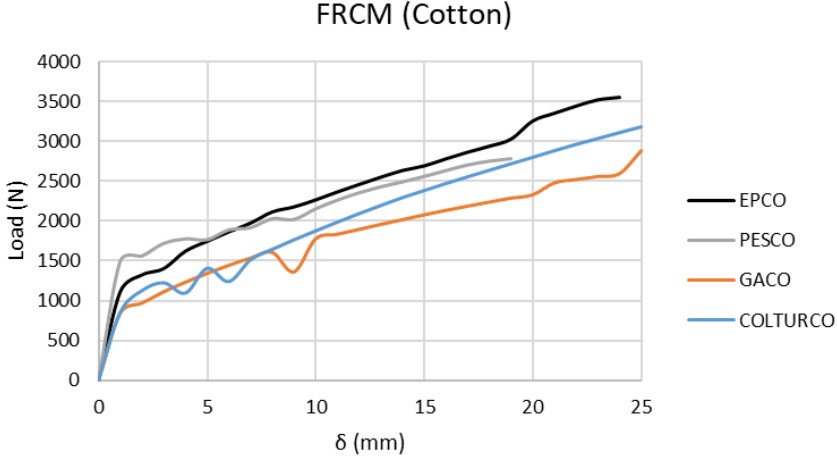

**Figure 9.** Load–Extensometer curves for cotton FRCM.

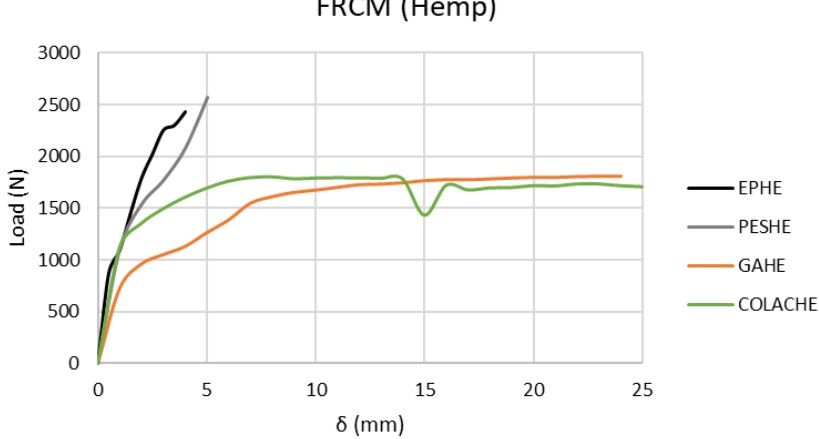

**Figure 10.** Load–Extensometer curves for hemp FRCM.

## 4. Discussion

### 4.1. Yarns Tensile Test

The deformation of the fibers was related to the unique properties of each fiber and the type of yarn used. The plaited yarns allowed the fibers to slide relatively from each other without debonding, presenting a plastic-like deformation, while spun yarns maintained the elastic deformation until failure, as shown in Figures 4 and 5.

First, the variation in the yarn properties when coating them (Tables 5 and 6) is discussed. Regarding the tensile strength, it suffered minor negative variations for cotton (under 10%) with the exception of colophony solved in turpentine, while hemp showed significant differences. Synthetic resins increased by more than 20% the hemp tensile strength while the natural resins decreased by more than 10%, with the exception of colophony solved in acetone, which increased by less than 10%.

The elastic modulus increased significantly with all except the GAU case with hemp yarns, while cotton suffered less significant variation, less than 20%. Mostly, in the cases where Young's modulus increased, the deformation was decreased for both yarns.

The impact of the coating of the yarns is seen clearer in hemp yarns. It was a spanned yarn, and the diameter was minor because the effect of the resin was greater. For cotton yarns, since it was a plaited yarn with a higher diameter, the area to coat was three times the hemp area, and the effect of the resin was lower, as the yarn maintained mostly the non-coated properties.

In order to continue the experimental campaign, two synthetic and two natural coated cases were selected: epoxy and polyester as synthetic, and Arabic gum and the colophony most suitable for each yarn (solved in acetone, COLAC, for hemp and solved in turpentine, COLTUR, for cotton).

### 4.2. Meshes Tensile Test

Epoxy resin is the most used resin to coat yarns and meshes for structural purposes, and it was taken as the reference when compared with the effect of the other resins (Tables 7 and 8).

As explained with the yarns, the hemp specimens showed more significant variations of mechanical properties than cotton ones.

Meshes showed 10% less stiffness than the corresponding yarns but reached a similar strength. It can be due to the load transmission between the yarns during the test, providing minor variations of the general properties.

For hemp, all coatings reduced their strength when compared with epoxy, and the variation in Young's modulus is less than 20%. The elongation was only increased using Arabic gum resin due to the flexibility of the resin. It allows the long fibers to conform to the yarns and slide without losing contact.

For cotton ones, which showed a non-linear response, the polyester resin was the only one that increased the strength while reducing the deformation at peak and maintaining a similar Young's modulus. Cotton yarns and meshes have both types of deformation, elastic and plastic, and the maximum strength was reached during plastic deformation, meaning that the effect of the polyester resin was more significant in that stage.

Arabic gum provided a higher deformation at the ultimate tensile strength, as it allowed the fibers to slide relatively from each other without losing contact.

The effect of the colophony was considered less significant than the Arabic gum due to not only the mechanical properties of the meshes but also their consistency. The meshes had to be handled carefully as the manipulation could damage the specimen.

### 4.3. FRCM Tensile Test

Regarding the FRCM specimens, they showed a two-stage behavior, where the first stage is stiffer, and after the cracking and load transmission, it is reduced until failing through slippage.

Hemp specimens have a more heterogenic behavior. Synthetic coating (polyester) provided higher strength values and failed through slippage or rupture of the mesh. On the

other hand, natural coatings presented a complete slippage of the fibers from the mortar matrix. For cotton ones, the results were more homogeneous, maintaining the failure modes depending on the source of the resin.

By observing Tables 9 and 10, the effect of the fiber reinforcement can be seen clearly during the first stage, where the specimen has not cracked yet. Even if it is considered that mainly the mortar was bearing the load, the slope was higher for hemp specimens when compared with the cotton cases.

For the second stage, where multicracking and sliding happened, polyester and natural coating reduced the slope with the exception of PESHE. It can be related to the failure mode of each case. Polyester-coated hemp specimens were the only ones that failed by mesh rupture, which means that the load transmission between the materials was correct (Figure 8, II).

Synthetic-coated specimens' failure consisted of the relative sliding of the mesh with the mortar, which resulted in cutting the mortar matrix (Figure 8, I), with the exception of PESHE. The anchoring system, a Clevis grip, where there was not any element that pressed the ends of the specimens while applying the load with low fiber–matrix compatibility, caused a relative displacement between the mesh and the matrix. The only case that had the expected failure was polyester-coated hemp FRCM.

Finally, the naturally coated specimens also showed a different failure mode, as the mesh completely slid from the mortar matrix. After the multicracking stage, the longitudinal yarns were not broken, they debonded completely from the matrix, and the remaining strength consisted of the friction between the fibers and the matrix. Which clearly states the low compatibility between natural-coated fibers and the matrix for the concentrations used in this research.

The surface of the fibers was smoother for synthetic coated fibers, probably due to the reticulation time. Liquids tend to form spherical shapes to reduce the superficial tensions [30]; therefore, the increase in the reticulation time allowed the natural resins to reshape homogeneously, reducing the roughness of the surface.

Furthermore, considering the epoxy-coated specimens as a reference, in Figures 9 and 10, there is a variation in the load and the different slopes of each case.

Polyester resin increased the tensile strength and stiffness of the specimens, while the Arabic gum increased the stiffness of the specimens but decreased their tensile strength. Finally, colophony increased the stiffness of the specimens and also increased the cotton specimen's strength but decreased the hemp ones. In general, all three coatings increased the stiffness when compared with epoxy coating as the epoxy coating is more flexible than the other ones.

Epoxy resin was the coating that provided more stiffness during the first stage in all cases except for PESCO specimens, where the polyester resin exceeded the stiffness of epoxy. Even if during the yarns and meshes tensile test, there are vegetal coatings that provide higher stiffness, in FRCM specimens, its stiffness decreased. It can be due to the deterioration of the coatings during the curing period because of the alkalinity environment of the mortar matrix.

For natural coatings, Arabic gum showed less compatibility with the matrix as the strength of the specimens decreased. It can be due to the fact that Arabic gum's solvent is distilled water, which maybe was more diluted during the curing period of the mortar. In the case of colophony, the effect is different depending on the type of fiber to coat. Probably due to the resistance of the fiber to the solvent (acetone), where cotton has more resistance.

For hemp specimens, the polyester resin showed higher compatibility with the fiber and the matrix because the specimens failed through the failure of the mesh. In the same case for cotton, since it is wider and plaited, the interphase was not good enough to transmit the necessary load to break the mesh.

A last empirical observation was the yarns' conditions after testing. The synthetic coated yarns still maintained the coating on their surface, but the natural coatings seemed

to have lost the superficial coating. It can be considered that it has been lost through chemical reactions with the fresh mortar or due the friction with the mortar matrix.

## 5. Analytical Model

The main failure observed was the slippage of the mesh, which led to two types of sliding. In the case of the synthetic resin, the mesh (as a unique element) slipped, cutting the mortar in two parts, while the natural coatings, with a lower interphase bond, slipped inside the mortar matrix, as shown in Figure 8.

One of the ACK model hypotheses consists of the assumption that the matrix–fiber bond is weak [4]. However, the failure mode presented by the model is the rupture of the mesh. In this research, where the main mode consisted of the shear failure of the contact plane or the slippage of the meshes inside the mortar matrix, it indicates that the mesh is not fully in traction.

In order to model this failure, a two-stage model approach was decided. The first stage consisted of the law of mixtures, commonly used in analytical composite models [19]. It combines Young's modulus of each material and its volumetric fraction.

$$E_{I,model} = E_m \cdot V_m + E_{mesh} \cdot V_{mesh} \tag{1}$$

where E stands for Young's modulus of the mortar and the mesh, and V stands for the volumetric fraction.

The law of mixtures is followed until the mortar's first crack, which can be calculated as follows:

$$\sigma_{mc,model} = (E_{I,\ model} \cdot \cdot \cdot \sigma_{mu})/E_m \tag{2}$$

where $\sigma_{mu}$ is the tensile strength of the mortar. The final deformation of the first stage is calculated as follows:

$$\varepsilon_{I,model} = \sigma_{mc,model}/E_{I,model} \tag{3}$$

The slope of the second stage consists of Hooke's law. It is combined with an effective coefficient ($\alpha$), defined by the type of fiber used on each specimen, that was tuned to fit the experimental curve.

$$E_{II,model} = E_{mesh} \cdot V_{mesh} \cdot \alpha \tag{4}$$

In order to calculate the model, $E_m = 8.92$ GPa [2] and $\sigma_{mu} = 2.26$ MPa were considered.

The results of the proposed model can be found in Table 11. The experimental and modeled slopes and the efficient coefficient were used. PESHE specimens were not modeled as their failure mode was not the one proposed by the model.

**Table 11.** Experimental and two-stage model comparison.

| Fiber | Coating | Two-Stage Model | | | | Experimental | |
|-------|---------|-----------------|--|--|--|--------------|--|
| | | $\sigma_{mc,model}$ (MPa) | $E_{I,model}$ (MPa) | $E_{II,model}$ (MPa) | $\alpha$ (-) | $E_I$ (MPa) | $E_{II}$ (MPa) |
| CO | EP | 2.18 | 8 613 | 14.29 | 0.20 | 111 | 9.93 |
| | PES | 2.18 | 8 605 | 12.62 | 0.20 | 76 | 9.80 |
| | GA | 2.18 | 8 617 | 15.10 | 0.20 | 84 | 7.23 |
| | COLTUR | 2.18 | 8 606 | 16.00 | 0.20 | 85 | 9.54 |
| HE | EP | 2.36 | 9 321 | 15.39 | 0.03 | 173 | 71.93 |
| | PES | - | - | - | - | - | - |
| | GA | 2.33 | 9 190 | 11.47 | 0.03 | 72 | 3.28 |
| | COLAC | 2.36 | 9 328 | 15.61 | 0.03 | 112 | 0.43 |

The curves comparing the FRCM experimental behaviour and the modelled behaviour can be found in Figures 11 and 12.

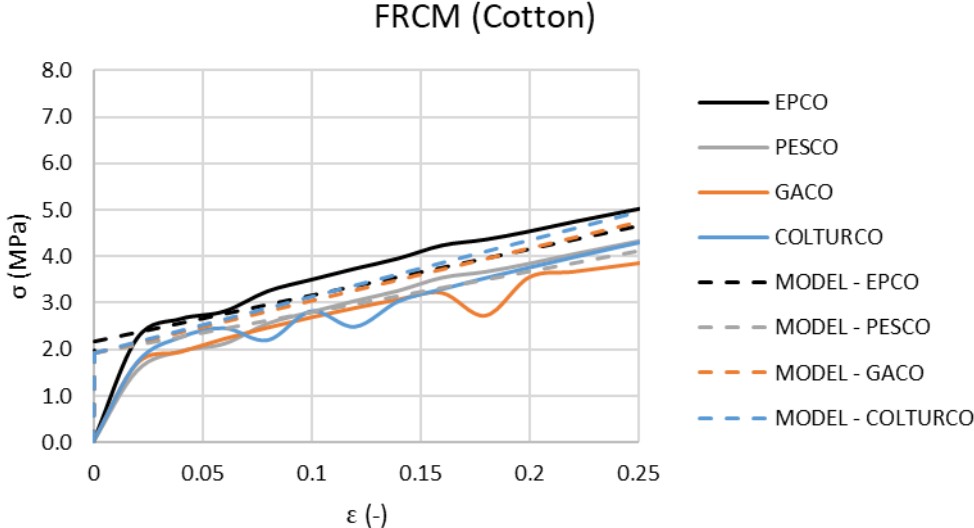

**Figure 11.** Stress–strain curves for cotton FRCM, analytical and experimental.

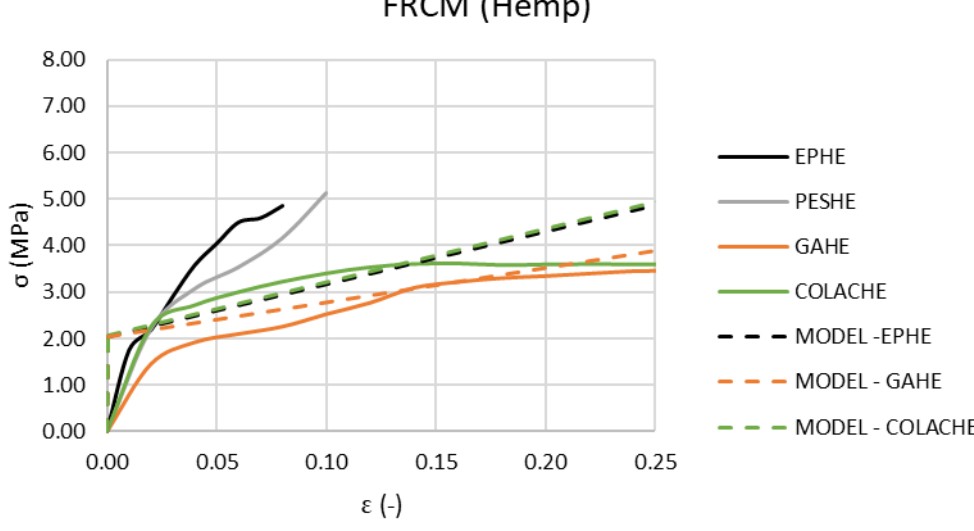

**Figure 12.** Stress–strain curves for hemp FRCM, analytical and experimental.

As in other studies, the law of mixtures on composite materials increased the stiffness of the first stage [4]. For the second stage, the cotton specimens modeled corresponded with the experimental curves. In the case of hemp, the models do not fit the experimental curves accurately; it can be due to the low perimeter of the hemp mesh with the matrix, which does not maintain the friction between the mesh and the matrix.

## 6. Conclusions

The experimental characterization of vegetal FRCM was performed successfully, obtaining the mechanical properties and failure modes of different vegetal FRCM combinations.

To conclude, the following lines show the conclusions of this research:

- The type of yarn used had an impact on the deformation behavior of the yarns. Plaited yarns presented plastic deformation as they had more capacity to slide between each other without fail.
- For yarns, the coatings had a more than 25% variation in the mechanical properties of hemp yarns. In cotton yarns, the effect was minor as the surface to coat was three times higher for cotton, and the yarn manufacture was different.

- Meshes are, in general, 10% less stiff than the corresponding yarns but reach a similar strength. It can be due to the fact that there is a load transmission between the yarns during the test, providing minor variations of the general properties.
- The coating of the fibers can completely modify the failure mode of the specimen, as natural coats consistently slide from the matrix, and synthetic coatings make specimens fail by mortar or mesh failure.
- Synthetic resins provided the roughest and strongest coating when compared with natural ones. It provided a higher mechanical interaction with the matrix, increasing the strength of the specimens.
- The residual coating of the composite yarns was maintained for synthetic-coated ones, but natural-coated ones seem to have been removed. It can be due to the chemical reaction that the coating may have with the fresh mortar or due to the friction while conducting the tensile test.
- The proposed two-stage model accurately represents the cotton specimens, while it was not correct for the hemp ones. It can be due to the heterogeneity of the experimental hemp results.

**Supplementary Materials:** The following supporting information can be downloaded at: https://www.mdpi.com/article/10.3390/app122412964/s1, Figure S1: Load-displacement curves for cotton yarns; Figure S2: Load-displacement curves for EPCO yarns; Figure S3: Load-displacement curves for PESCO yarns; Figure S4: Load-displacement curves for GACO yarns; Figure S5: Load-displacement curves for GAUCO yarns; Figure S6: Load-displacement curves for COLTURCO yarns; Figure S7: Load-displacement curves for COLACCO yarns; Figure S8: Load-displacement curves for hemp yarns; Figure S9: Load-displacement curves for EPHE yarns; Figure S10: Load-displacement curves for PESHE yarns; Figure S11: Load-displacement curves for GAHE yarns; Figure S12: Load-displacement curves for GAUHE yarns; Figure S13: Load-displacement curves for COLTURHE yarns; Figure S14: Load-displacement curves for COLACHE yarns; Figure S15: Load-displacement curves for EPCO meshes; Figure S16: Load-displacement curves for PESCO meshes; Figure S17: Load-displacement curves for GACO meshes; Figure S18: Load-displacement curves for COLTURCO meshes; Figure S19: Load-displacement curves for EPHE meshes; Figure S20: Load-displacement curves for PESHE meshes; Figure S21: Load-displacement curves for GAHE meshes; Figure S22: Load-displacement curves for COLACHE meshes; Figure S23: Load-displacement curves for EPCO FRCMs; Figure S24: Load-displacement curves for PESCO FRCMs; Figure S25: Load-displacement curves for GACO FRCMs; Figure S26: Load-displacement curves for COLTURCO FRCMs; Figure S27: Load-displacement curves for EPHE FRCMs; Figure S28: Load-displacement curves for PESHE FRCMs; Figure S29: Load-displacement curves for GAHE FRCMs; Figure S30: Load-displacement curves for COLACHE FRCMs.

**Author Contributions:** Conceptualization, L.M. and V.M.; methodology, L.M. and B.M.; software, B.M.; validation, E.B.-M., B.M. and L.G.; formal analysis, V.M.; investigation, V.M.; resources, L.G.; data curation, V.M.; writing—original draft preparation, V.M.; writing—review and editing, V.M., E.B.-M. and L.G.; visualization, L.M.; supervision, L.G. and E.B.-M.; project administration, L.G.; funding acquisition, L.G. All authors have read and agreed to the published version of the manuscript.

**Funding:** The authors gratefully acknowledge partial financial support from AZVI, S.A. company through the REDUVE contract C-11402. The corresponding author is partially granted through the FPI-UPC scholarship, 2019. The second author is partially granted through the FPU-UPC scholarship, 2021. The fourth author is a Serra Húnter fellow.

**Institutional Review Board Statement:** Not applicable.

**Informed Consent Statement:** Not applicable.

**Data Availability Statement:** Not applicable.

**Conflicts of Interest:** The authors declare no conflict of interest.

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
