# Peer review of "Vegetal-FRCM Failure under Partial Interaction Mechanism"

_applsci, doi:10.3390/app122412964_

Round 1

Reviewer 1 Report

The manuscript focuses on an experimental investigation on vegetal-FRCM, by using different types of coatings and fibers. A bilinear law is also proposed to analytically represent their experimental behavior The topic is certainly interesting; however, the quality of the manuscript, and especially of tables and graphs, should be deeply improved before publication.

Some of the Tables are incomplete; for example:

In Table 4: COLAC coating is not reported. Moreover, the reported denominations have not yet been defined in the text when the Table is inserted and consequently they are not so clear.

In Table 5, with reference to hemp yarns, the line relative to COLAC coating is missing. Moreover, the adopted symbols should be defined somewhere in the text or with footnotes under the table. Some more details about the procedure followed to estimate the elastic modulus should be also provided.

The data reported in Table 7 sometimes do not correspond with the curves reported in Figures 8 and 9: for example, 3 curves for cotton end for delta=25 mm while delta peak in the table range between 20.51 and 23.66. Moreover, for hemp, delta,peak for COLAC is quite different from that reported in the corresponding graph; the same apply for the peak load for GAHE. Why a lower sigma peak is associated to higher values of Fpeak? Please explain.

Some little discrepancies are also present in Table 6 with respect to Figures 4 and 5, please check. The failure modes should be better discussed so to make clearly understandable the difference between “sliding of the mesh” and “total sliding of the mesh from the mortar matrix”. A Figure reporting the failure modes should be added.

Table 8: what is the difference between the reported sigma,mc and sigma,mc,model (if any?).

In Figure 5, the chart legend is missing, so the graph is completely unreadable. What do you mean with estensometer on the x-axis? Maybe displacement? Please clarify and correct also in the other graphs.

In Figures 10 and 11: it is not clear why the y-axis legend is tau and not sigma, please explain

The text of Section 3.2 is a simple copy and paste from Section 3.1, while it should have been adapted to the mesh case. At this point of the reading, it is not so clear why only some specimens are reported in Table 6.

In Section 3.3, the reference to the Figures 6-8 is wrong and the Figures related to the failure modes are missing.

The text at lines 306-315 is not so clear with reference to Figures 8 and 9: if epoxy coating is the reference one, the only case showing higher stiffness and strength is PES for cotton fibers. Please explain.

Line 335: Please explain how parameter alpha has been calibrated. It is stated that sigma,mu = 2.26 MPa is assumed from the experimental campaign… which one? In Table 3, the tensile strength of mortar is declared equal to 2.9 MPa.

Other required changes to improve the quality of the work are listed below:

-        English should be deeply revised to remove typos and grammar errors. Some sentences should be corrected or rephrased to enhance the readability of the text (e.g. lines 15-16, 80-81, 104-105, 195-196, 235-236, 254-255, 296-297, 317, 338-339, 349-350)

Sometimes the dot at the end of a sentence is missing or misplaced (see, e.g., lines 155, 175, 192, 315, 364, etc..)

Replace: cementitic with cementitious, set-ups with setup, tensed with in tension, breakage with failure or rupture

-        In Tables, the decimal separator is sometimes a dot, and other times a comma. Please, use the same symbol.

-        Most of the Figures can be enlarged. The title and the caption of Figure 9 should be corrected.

-        In the Introduction, when discussing the tri-linear model usually adopted to model FRCM behavior – that in some cases may be not correct, other references should be added, like for example: M. Butler et al., Durability of textile reinforced concrete made with AR glass fibre: effect of matrix composition, Mater Struct, 43 (2010), pp. 1351-1368; G. Carozzi et al. Mechanical properties and debonding strength of Fabric Reinforced Cementitious Matrix (FRCM) systems for masonry strengthening, Composites Part B: Engineering, 70 (2015), pp. 215-230; ; Bernardi et al. (2016). A non-linear constitutive relation for the analysis of FRCM elements. Procedia Structural Integrity, 2, 2674-2681; Minafò, G et al. (2018). Experimental investigation on the effect of mortar grade on the compressive behaviour of FRCM confined masonry columns. Composites Part B: Engineering, 146, 1-12.

-        Lines 12-15: this sentence is not completely clear and should be better discussed. In fact, also in ACK model the hypotheses are that the matrix-fibre bond is weak, and that once the matrix and the fibre debond, a pure frictional shear stress rules the matrix-fibre interface behavior, assumed as constant along the debonded interface.

-        Lines 226-228: a Figure should be added for showing the described mechanism. The sentence should be corrected because “plaited yarns” is repeated twice.

-        Line 304: the reference to Figure 6 is wrong. Line 325: the reference to Figures 7 and 8 is wrong

-        Check reference 24.

Author Response

Dear reviewer #1,

Thank you for taking the time to review and comment upon our manuscript, 2028753 - Vegetal-FRCM failure under partial interaction mechanism.

Your contributions were very constructive and all suggestions have been taken into account in our revision. The changes on the document are highlighted by colors, in red and crossed out the deleted parts and in green the added new parts.

Two main points:

  1. According your comments, we noticed that it would be worth to add some supplementary material with all the original plots of all tests. We have not previously included them because we didn’t want to enlarge the paper size; but the choice of supplementary data satisfies your proper observation about the lack of data and we keep the length of the paper.
  2. In order to help you to follow the scientific changes of the current manuscript we leave the English review for a post-review. We have agreed with the editor that if the paper is accepted, we will proceed with an English language review using the MPDI service. Sorry for this inconvenience, but we are sure that you can understand it and this procedure provides a cleaner improvement track.

The following table will answer each one of the reviewer’s comments.

#

Reviewer comment

Answer

1

In Table 4: COLAC coating is not reported. Moreover, the reported denominations have not yet been defined in the text when the Table is inserted and consequently, they are not so clear.

The reviewer was right. The nomenclature wasn’t presented before. Therefore, the missing row was added and also the meaning of the nomenclature in Table 4.

2

In Table 5, with reference to hemp yarns, the line relative to COLAC coating is missing. Moreover, the adopted symbols should be defined somewhere in the text or with footnotes under the table. Some more details about the procedure followed to estimate the elastic modulus should be also provided.

The reviewer was right. The line has been added in Table 5 and the definition of the symbols were added to the text (lines 196-203).

3

The data reported in Table 7 sometimes do not correspond with the curves reported in Figures 8 and 9: for example, 3 curves for cotton end for delta=25 mm while delta peak in the table range between 20.51 and 23.66. Moreover, for hemp, delta,peak for COLAC is quite different from that reported in the corresponding graph; the same apply for the peak load for GAHE.

That is because the graphs have been calculated as the average load per displacement.

It was taken a fix displacement p.e. 1mm, and the three loads were used to calculate the average.

Therefore, the maximum or the end of the test doesn’t exactly match with the tables.

To be clearer, 3 modifications were taken into account:

-       Put all three peak loads and displacements for every case in the tables, keeping also the averages.

-       It was stated below the corresponding figures that it is an average curve calculated using the average loads per displacement.

-       It was added to Supplementary materials all the curves per each case.

4

Why a lower sigma peak is associated to higher values of Fpeak? Please explain.

Because the area of the meshes are different. The area of the cotton mesh is 21.21 mm2 and for hemp ones 6.28 mm2. Therefore, the cotton that bears more load, has a smaller strength.

The explanation was added to the paper.

5

Some little discrepancies are also present in Table 6 with respect to Figures 4 and 5, please check.

Answered in comment #3.

6

The failure modes should be better discussed so to make clearly understandable the difference between “sliding of the mesh” and “total sliding of the mesh from the mortar matrix”. A Figure reporting the failure modes should be added.

The failure modes have been defined in text and a figure was added for more understanding. (Lines 247-249, Figure 8).

7

Table 8: what is the difference between the reported sigma,mc and sigma,mc,model (if any?).

Thank you for the noticing, it was the same value. It has been homogenized to sigma,mc,model. (Table 11)

8

In Figure 5, the chart legend is missing, so the graph is completely unreadable. What do you mean with extensometer on the x-axis? Maybe displacement? Please clarify and correct also in the other graphs.

That happened due the editing to place the format. It has been corrected. And now figure 4 and 5 have the labels.

It is actually the extensometer displacement. It has been homogenized to delta, and explained in text that was the displacement of the extensometer. (Lines 197-198, 224-225, 245-246)

9

In Figures 10 and 11: it is not clear why the y-axis legend is tau and not sigma, please explain.

It has been changed to sigma. It was a T commonly used in Spanish. (Figures 11 and 12)

10

The text of Section 3.2 is a simple copy and paste from Section 3.1, while it should have been adapted to the mesh case. At this point of the reading, it is not so clear why only some specimens are reported in Table 6.

Thank you for noticing it, it was re-written for the meshes.

 After the yarns’ figures it was stated the reasoning of the reduction of the cases for the following experimental campaign. (Lines 214-218)

11

In Section 3.3, the reference to the Figures 6-8 is wrong and the Figures related to the failure modes are missing.

We agree. The references have been changed and the failure modes’ figure added. (Lines 253, Figure 8).

12

The text at lines 306-315 is not so clear with reference to Figures 8 and 9: if epoxy coating is the reference one, the only case showing higher stiffness and strength is PES for cotton fibers. Please explain.

The reviewer was right. The paragraph wasn’t clear. Therefore, it was substituted by two paragraphs (lines 354-359, 366-369).

13

Line 335: Please explain how parameter alpha has been calibrated.

The parameter alpha was tuned to fit the experimental curve. (Line 393-394)

14

It is stated that sigma,mu = 2.26 MPa is assumed from the experimental campaign… which one? In Table 3, the tensile strength of mortar is declared equal to 2.9 MPa.

2.9 MPa corresponded to previous values from previous experimental campaigns that now have been updated with the ones of the current research after test repetition. Table 3 was not updated and now it has been.

15

English should be deeply revised to remove typos and grammar errors. Some sentences should be corrected or rephrased to enhance the readability of the text (e.g. lines 15-16, 80-81, 104-105, 195-196, 235-236, 254-255, 296-297, 317, 338-339, 349-350)

The lines has been changed and corrected to enhance readability.

16

Sometimes the dot at the end of a sentence is missing or misplaced (see, e.g., lines 155, 175, 192, 315, 364, etc…)

The punctuation has been revised and corrected.

17

In Tables, the decimal separator is sometimes a dot, and other times a comma. Please, use the same symbol.

The decimal separator has been corrected to dot in every table.

18

Most of the Figures can be enlarged. The title and the caption of Figure 9 should be corrected.

The figures have been enlarged and Figure 12’s (before Figure 9) title corrected.

19

In the Introduction, when discussing the tri-linear model usually adopted to model FRCM behavior – that in some cases may be not correct, other references should be added, like for example:

We found the references quite interesting and added them to the introduction.

20

Lines 12-15: this sentence is not completely clear and should be better discussed. In fact, also in ACK model the hypotheses are that the matrix-fibre bond is weak, and that once the matrix and the fibre debond, a pure frictional shear stress rules the matrix-fibre interface behavior, assumed as constant along the debonded interface.

The sentence has been modified to be clearer.

We agree that one of the hypotheses of ACK is the weak matrix-fiber bond. Still, the failure mechanism that ACK suggests is the failure of the mesh, as it elongates during the third stage.

Here what we’ve found is that the mechanical lock of the mesh with the mortar is extremely bad, and its interphase provokes a plane sliding, cutting the mortar, which suggests that the meshes haven’t been fully in traction.

Lines now 14-15 have been changed. And an explanation in lines 380-384 has been added.

21

Lines 226-228: a Figure should be added for showing the described mechanism. The sentence should be corrected because “plaited yarns” is repeated twice.

The second plaited was supposed to be spun, and the reference to the experimental curves have been added. (line 270)

22

Line 304: the reference to Figure 6 is wrong. Line 325: the reference to Figures 7 and 8 is wrong.

The references have been checked and corrected. (line 349)

The reference to figures 7 and 8 has been changed too. (line 380).

23

Check reference 24.

The reference 29 (before 24) has been corrected.

Thank you again for your thoughtful comments.

Sincerely,

Virginia Mendizabal et al.

Reviewer 2 Report

The manuscript entitled: "Vegetal-FRCM failure under partial interaction mechanism" proposes a two-stage analytical model aiming at increasing the knowledge about fiber-reinforced systems that employ natural fibers.  The focus of the work is on the performances of these systems when a partial interaction arises. The authors are presenting an extended experimental campaign developed comparing the performance of different synthetic-coated and natural-coated vegetal FRCMs. The campaign aims then at showing the complete characterization of the materials starting from the yarn to the final FRCM composite material. 

The paper is well-organized and clear to the reader, it is also providing clear results and discussion of the main points treated. 

Nevertheless, I'm suggesting some minor improvements useful to increase the quality of the manuscript.

I would improve the quality of the figures in general: Figures 4 and 5 have some problems with the labels.

Some minor improvements in the English language should be addressed, such as (not an exhaustive list):

line 248: epoxy resin IS the most...

line 249: compared instead of comparing

line 256: their instead of its and, compared instead of comparing

line 260: a non-linear

line 273: transmission, it is reduced

Please check all the paper text with grammar spell check.

Author Response

Dear reviewer #2,

Thank you for taking the time to review and comment upon our manuscript, 2028753- Vegetal-FRCM failure under partial interaction mechanism.

All the suggestions have been considered into our revision. Additionally some supplementary data has been included with all tests plots.

The changes on the document are highlighted by colors, in red and crossed out the deleted parts and in green the added new parts.

Moreover, in order to help you to follow the scientific changes of the current manuscript we leave the English review for a post-review. We have agreed with the editor that if the paper is accepted, we will proceed with an English language review using the MPDI service. Sorry for this inconvenience, but we are sure that you can understand it and this procedure provides a cleaner improvement track.

Thank you again for your thoughtful comments.

Sincerely,

Virginia Mendizabal et al.
